# Offshore Measurements and Numerical Validation of the Mooring Forces on a 1:5 Scale Buoy

**Jens Engström** [1,*], **Zahra Shahroozi** [1], **Eirini Katsidoniotaki** [1,2], **Charitini Stavropoulou** [1], **Pär Johannesson** [3] **and Malin Göteman** [1,2]

1   Department of Electrical Engineering, The Ångström Laboratory, Uppsala University, 752 37 Uppsala, Sweden
2   Centre of Natural Hazards and Disaster Science (CNDS), Villavägen 16, 752 36 Uppsala, Sweden
3   RISE Research Institutes of Sweden, Department of Applied Mechanics, Gibraltargatan 35, 412 79 Gothenburg, Sweden
*   Correspondence: jens.engstrom@angstrom.uu.se

**Abstract:** Wave energy conversion is a renewable energy technology with a promising potential. Although it has been developed for more than 200 years, the technology is still far from mature. The survivability in extreme weather conditions is a key parameter halting its development. We present here results from two weeks of measurement with a force measurement buoy deployed at Uppsala University's test site for wave energy research at the west coast of Sweden. The collected data have been used to investigate the reliability for two typical numerical wave energy converter models: one low fidelity model based on linear wave theory and one high fidelity Reynolds-Averaged Navier–Stokes model. The line force data is also analysed by extreme value theory using the peak-over-threshold method to study the statistical distribution of extreme forces and to predict the return period. The high fidelity model shows rather good agreement for the smaller waves, but overestimates the forces for larger waves, which can be attributed to uncertainties related to field measurements and numerical modelling uncertainties. The peak-over-threshold method gives a rather satisfying result for this data set. A significant deviation is observed in the measured force for sea states with the same significant wave height. This indicates that it will be difficult to calculate the force based on the significant wave height only, which points out the importance of more offshore experiments.

**Keywords:** wave energy conversion; point absorber; line force; offshore measurements

## 1. Introduction

For wave energy to become an economically viable energy source, the technology has to be designed for the most common wave, but withstand power levels during storms and other extreme conditions. Even if extreme waves are rare, a single occurrence may be disastrous if it leads to a failure of the device. The loading on a wave energy converter (WEC) can be 100 times higher than the average loading in extreme weather conditions [1], which provides a significant challenge to the structural design and consequentially capital costs. Wave load impact has long been a subject of study for ships, platforms and other offshore structures, but poses a new challenge for wave energy systems due to the different scales and dynamics involved.

When studying the dynamics in steep waves, nonlinear phenomena become important. Complex free surface phenomena such as wave breaking, which have a large impact on the survivability of offshore structures, can only be studied with full computational fluid dynamics (CFD) methods [2]. In general, extreme waves are sporadic events embedded within a random sea state, making their prediction and reproduction difficult. In addition, the specific wave group combinations responsible for the most severe loads on dynamic structures remain unclear and can only be found by testing a range of conditions [2] both numerically and experimentally. Given the computation or measurements of load peaks in different wave conditions, the distribution of loads can be identified for a certain sea

state, and the peaks-over-threshold method can be used to estimate the long-term force distribution [3].

Due to a limited number of experiments with wave energy converters offshore and the fact that device developers conducting those experiments are often enterprises that seldom share data, there are a limited amount of results from offshore experiments in the literature. However, there are a few device developers that have published data from offshore experiments. Power production data have been presented by Uppsala University [4], Wave Star [5], Pelamis [6], Mutriku [7], and Ocean Energy [8]. A few experimental studies have been performed to study extreme forces on wave energy converters. Physical experiments as well as numerical simulations of wave loads on different point absorber WECs were reported in [9,10], and offshore measurements of line forces were presented in [11]. A combined approach of numerical modelling and experimental tests to study the failure of shackles was presented in [12], and in [13] it was shown that snap loads could lead to failure in mooring lines. Snap loads were also studied experimentally in [14], and large differences in peak load magnitude and propagation speed were shown. In [15], wave loads on a point absorber model in extreme waves as well as more operational irregular waves were measured, and the results showed a significant spread in the wave loads and that the generator damping plays a major role in reducing the wave impact. These studies show that the magnitude of the extreme forces is largely influenced by the design and dimensions of the device in question, as well as by the dynamics and specific time-history of the wave event. Additionally, since experiments are expensive and difficult to carry out, numerical simulations are indispensable tools, and state-of-the-art CFD software is becoming increasingly mature and reliable, in particular with access to high performing computer clusters. Various CFD models have been developed [16–18] and some validated to experimental data from wave tank tests of scale models of point absorbing WECs [19]. However, numerical treatment of extreme loads in dynamical offshore systems remain a challenge. For example, overshoots due to the discontinuous nature of a shock can give an overestimated peak load unless slope limiters are used [20].

To fully understand the loading on a WEC of point absorber type in real operational conditions, full-scale experiments are needed. However, a scale model subjected to the same environmental conditions may provide data that are not that far from the real situation. The work in this paper presents one step towards that goal, and the data are used in several numerical models for extreme force calculations and predictions. In this paper, we present results from two weeks of force measurements on a 1 m diameter buoy at an offshore test site. The site is situated at Lysekil on the Swedish west coast and has been used to test wave energy converters since 2006 [21,22]. The size of the buoy yields a 1:5 scale relation to Uppsala University's point absorber wave energy converter. This is the main novelty of this paper: *actual offshore data is presented and analysed on extreme force measurements on a WEC model.* Since published literature on realistic offshore measurements relevant for wave energy is extremely scarce, this contributes to closing knowledge gaps needed to design WECs able to survive in harsh offshore conditions.

In addition to the experimental results, we present numerical results using two models: a low-fidelity linear potential flow model, and a high-fidelity CFD model. However, this is not a validation of numerical models. Instead, we compare two often used numerical models on both ends of the fidelity scale to offshore data to see how well these models reproduce the true environment. The line force data is also analysed by extreme value theory using the peak-over-threshold method to study the statistical distribution of extreme forces and to predict the return period, i.e., how often these extreme forces are expected. Section 2 presents the experimental set up and test site followed by a presentation of the numerical models. The remainder of the paper is structured as follows. In Section 3, the results are presented, and the paper ends with discussion and conclusions in Sections 4 and 5.

## 2. Method

### 2.1. Experiment

The experimental setup consists of a floating cylindrical buoy with 1 m diameter. The top and bottom of the buoy are hemispherical, see Figure 1. The size of the buoy is chosen to scale with 1:5 to a full-scale wave energy converter of the type developed at Uppsala University. The buoy has a closed force measurement system with an Applied Measurement DDEN in line load cell at the bottom of the buoy. Inside the buoy, there is one 12 V and 75 Ah battery and a MSR 145 A data logger, both connected to the load cell. The buoy is also equipped with 20 kg ballast to equalize the mass distribution of the buoy. The data logger was programmed to measure force with sampling frequency of 64 Hz for one month. The buoy was connected to a 1000 kg foundation via a UNIMER mooring compensator (rubber spring). The rubber spring was used to increase the survivability of the system. A similar setup was installed in 2017, but without a rubber spring and the line broke in a severe storm. The rubber spring has a spring constant of 5000 N/m. The set-up was deployed at a water depth of 25 m, and the line length was chosen to give a slack of approximately 0.64 m. The line was a 3-strand, 24 mm polyester silk rope. The top of the load cell is connected to the buoy via a shackle, and the bottom of the load cell is tied to the mooring line. The force measurement buoy was deployed on 10 May 2019 at 11:00 a.m. and retrieved on 11 June 2019. The cable to the load cell broke after 16 days. Details of the experiment are given in Table 1.

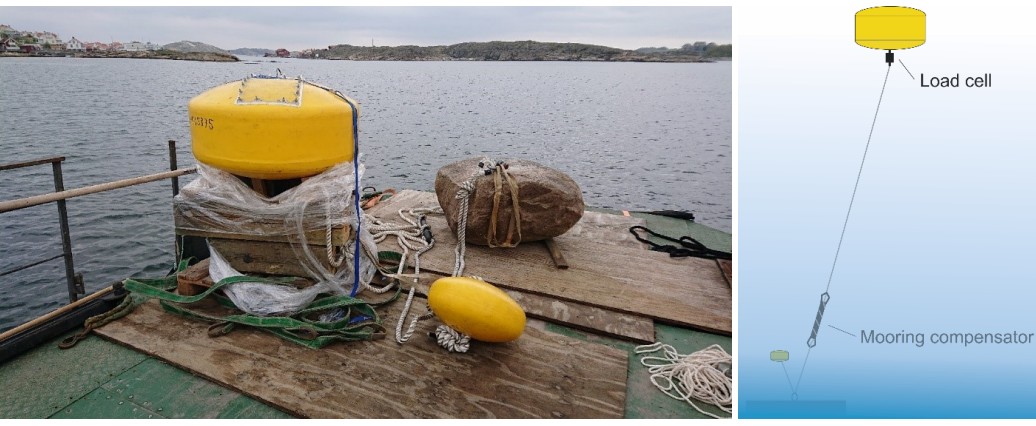

**Figure 1. Left** picture shows the experimental setup just before deployment. The **right** figure shows a sketch of the setup. The small buoy at the bottom is used to lift the shackle connecting the line and foundation in order to avoid wear on the line.

**Table 1.** Details of the experimental set up.

| Parameter | Value | Unit | Parameter | Value | Unit |
|---|---|---|---|---|---|
| Buoy diameter | 1 | m | Buoy volume | 0.35 | m$^3$ |
| Buoy height | 0.55 | m | Water depth | 25 | m |
| Buoy draft | 0.15 | m | Spring constant | 5000 | N/m |
| Buoy mass | 81 | kg | Mass of foundation | 1000 | kg |

The force data are divided into 30 min bins in the same way as the wave data to correspond to a certain sea state.

The force measurement buoy was deployed in calm conditions. Then followed a period of 2 days with normal waves ($H_s = 1 - 1.5$ m) for the site. After that, there was a long period with very calm conditions before a period with higher waves ($H_s < 2.5$ m) when the cable to the load cell broke. During the last period, one wave with an amplitude

of 2.4 m was measured. There was some data drift during the first days in the force data. To handle this, the mean value for each 30 min is subtracted from the original value.

### 2.2. Test Site

The experiment was carried out at Uppsala University's test site for wave energy conversion, which is located 2 km offshore just south of the town Lysekil on the Swedish west coast, situated between a northern marker (58°11′850″ N 11°22′460″ E) and a southern marker (58°11′630″ N 11°22′460″ E). The site is a near shore location, sheltered by small islands to the north and by the small islet of Klammerskäret to the south. To the west/south west, the site is open to Skagerrak and the North Sea. The seabed is fairly level with an average depth of 25 m. Variations in water level due to tides and air pressure variations for the time of the experiment are presented in Figure 2.

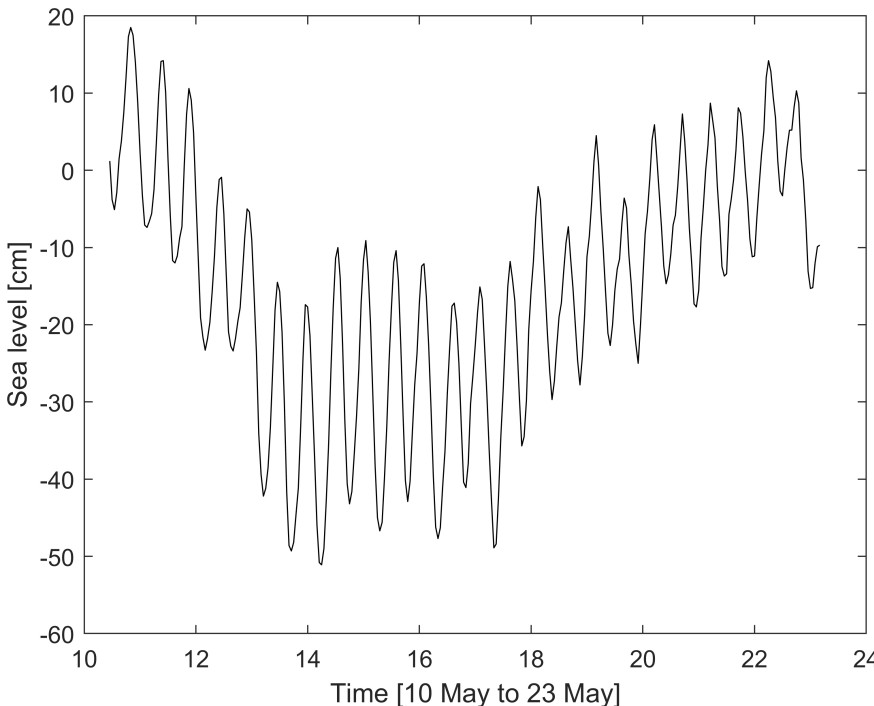

**Figure 2.** Water level variations for the period of the experiment. Data obtained from Sveriges meteorologiska och hydrologiska institut's measurement station Brofjorden (id:35109) located 16 km from the test site. Data are measured values.

As can be seen from Figure 2, the water level variations are very small and will have no major implications on the experimental data. Uppsala University has a Datawell Waverider buoy deployed at the site, approximately 100 m from the force measurement buoy, and wave data were collected continuously during the time of the experiment. The test site has a rather mild wave climate; 44% of the annual energy flux occurs for sea states with an energy period $T_e$ in the interval 4–7 s and a significant wave height $H_s$ in the interval 1–3 m [21]. The wave elevation measurements are divided into 30 min bins and the significant wave height $H_s$ and energy period $T_e$ are calculated for each bin. The significant wave height at the test site for the time period of the experiment is presented in Figure 3. The short discontinuity in the figure at approximately 5 days is due to erroneous data that has been removed. The Division for Electricity at Uppsala University has since 2004 developed a wave energy conversion technology of point absorber type with direct-driven linear generator as power take off (PTO) [22]. Twelve full scale wave energy converters and two marine substations have been deployed at the site.

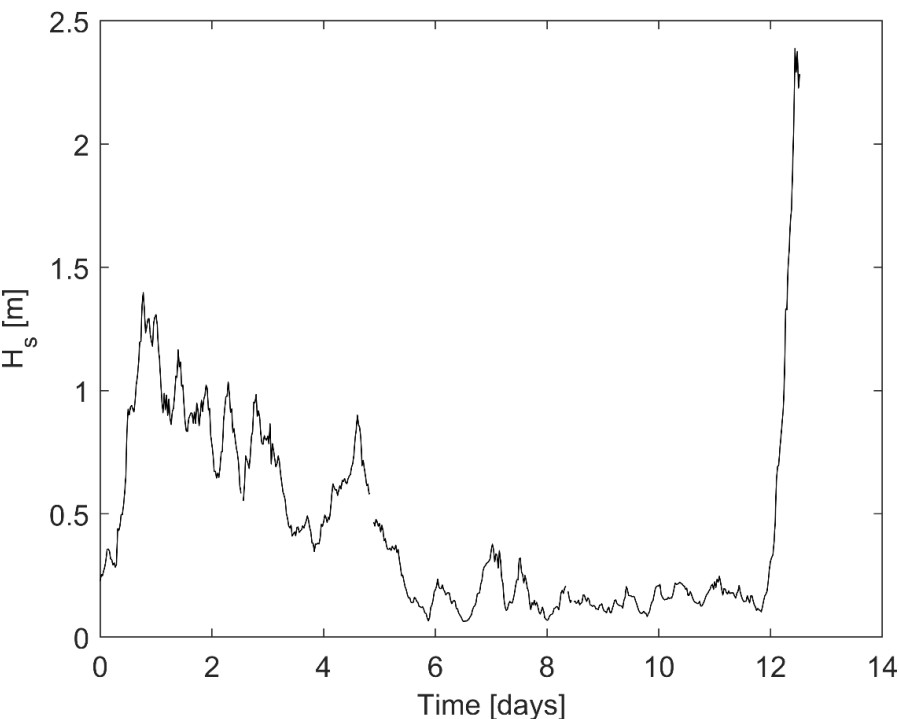

**Figure 3.** Significant wave height for the period of the experiment. Data measured at the test site by Uppsala University's wave measurement buoy.

### 2.3. Numerical Modelling

Two numerical methods are used in the paper to model the line forces—one simple linear model with a low computational cost, to allow for fast simulations of many sea states, and one advanced non-linear and viscous CFD model.

### 2.3.1. Linear Model

In the linear potential flow model, several crude approximations are done both for the fluid and the dynamics of the body. Viscosity and rotation are neglected, and the fluid is assumed to be incompressible, with small amplitude waves, such that the governing equations reduce to the standard Laplace equation and the linear boundary constraints on the free surface and the rigid body surfaces. Instead of the actual geometry of the buoy (shown in Figure 1), a cylinder buoy has been modelled. The cylinder has the same radius $R = 0.5$ m, mass $m = 81$ kg and the same submerged volume as the actual buoy, which is provided by draft $d = 0.10$ m. In addition, the buoy is assumed to move only in heave. The equations of motion for the surface buoy are given by Newton's second law,

$$m\ddot{z}(t) = F_{\text{exc}}(t) + F_{\text{rad}}(t) + F_{\text{stat}}(t) + F_{\text{spring}}(t) - F_{mg}, \tag{1}$$

where $z(t)$ is the vertical position of the buoy, $F_{\text{exc}}(t)$ and $F_{\text{rad}}(t)$ are the hydrodynamical excitation and radiation forces, and $F_{\text{stat}}(t) = -\rho\pi R^2(z(t) - d)$ is the hydrostatical restoring force. $F_{\text{spring}} = -kz(t)$ is the spring force of the rubber spring mooring system. A constant value of $k = 5000$ N/m has been used for the simulations in non-slack conditions, whereas during the experiment the spring constant was dependent on the applied force and varied between roughly 4700 N/m and 5200 N/m. Finally, $F_{mg}$ is the total weight of the buoy, given as $F_{mg} = m \cdot g$, where $g = 9.81$ m/s² is the gravitational acceleration. The hydrodynamical force coefficients are computed using the boundary element method software WAMIT and the equations of motion are solved in the time domain, where convolution terms appear in the computation of the excitation and radiation forces. Since the numerical scheme assumes that the float is moving only in heave, the convolution integrals can be easily computed

using direct numerical integration, implemented in MATLAB. After the equations of motion have been solved, the resulting vertical displacement of the buoy is used to compute the force due to the connection line between the buoy and the anchor point at the seabed, for the full duration of the sea state, as $F_{\text{line}}(t) = -kz(t)$.

The incident irregular waves that have been measured at the offshore site are used as inputs to the model. However, the buoy was deployed with a mooring line that was 0.64 m longer than the actual water depth, hence the phenomenon of a slack line can occur. Waves with a wave height below the superfluous line length will only stretch the line to a non-slack condition, but the mooring spring will only be extended in waves with higher wave height than 0.64 m. To implement this in the numerical MATLAB model, we used a simple *if* statement, so the spring force is active only if the vertical displacement of the buoy is greater than 0.64 m.

### 2.3.2. CFD Model

To complement the very simple yet fast linear potential flow model, high-fidelity CFD simulations have been carried out, on the same buoy geometry as for the linear model, to study the behaviour of the system, using the open source software OpenFOAM v1906. The *overInterDyMFoam* solver is used that is designed to solve Reynolds-Averaged Navier–Stokes for two incompressible, isothermal immiscible fluids (i.e., air and water) using the volume of fluid approach to capture the interface between air and water at the free surface [23]. This solver allows the overset mesh implementation for capturing the body's motion which has been previously applied by the authors in [18,24]. The overset method uses disconnected meshes to discretize the flow domain, the governing partial differential equations are solved in all the mesh regions and the information passes from a region to the other through interpolation. The pressure-velocity coupling is solved using the PIMPLE algorithm. Adjustable time step follows the Courant–Friedrichs–Lewy (CLF) condition with maximum CLF = 0.25. As it is outlined in [25], small variations in the results are noticed for CLF < 0.5. Turbulent effects become important in the real applications, therefore, turbulent flow is considered in this work. The Keulegan–Carpenter (KC) number determines the relation between the drag and inertia forces (KC = $\pi H/D$, $H$ and $D$ is the wave height and buoy diameter, respectively). In this application, KC = 5–14 (Table 2), inferring that the drag forces should not be neglected. As the most applied turbulence model in the WEC literature, the $k - \omega$ *SST* model is utilized [26]. The numerical simulations have been carried out in the same scale as the actual offshore experiments, i.e., 1:5. Each simulation completes 10 wave cycles.

**Table 2.** Characteristics of the four sea states investigated as equivalent regular waves using CFD.

| Sea State | $H_s$ [m] | $T_p$ [s] | $H$ [m] | $T$ [s] | KC |
|:---:|:---:|:---:|:---:|:---:|:---:|
| # 1 | 0.90 | 5.45 | 1.71 | 5.45 | 5.37 |
| # 2 | 1.02 | 7.63 | 1.95 | 7.63 | 6.12 |
| # 3 | 1.40 | 5.74 | 2.66 | 5.74 | 8.36 |
| # 4 | 2.38 | 8.00 | 4.52 | 8.00 | 14.0 |

The *sixDoFRigidBodyMotion* library, available in OpenFOAM, defines the translational and rotational body motion. In particular, the equation of motion (Equation (1)), as defined by Newton's second law, is solved based on the hydrodynamic force/torque and the external forces acting on the body (i.e., gravity, restraints). The solution of the governing RANS equations provides the hydrodynamic force, $F_{\text{h}}$, and torque, $M_{\text{h}}$, due to the wave interaction:

$$F_{\mathrm{h}} = \int_A (p\mathbf{n} + \tau)dS$$
$$M_{\mathrm{h}} = \int_A \mathbf{r} \times (p\mathbf{n} + \tau)dS \qquad (2)$$

where $p$ is the total pressure, $\mathbf{n}$ is the unit normal vector, $\tau$ is the shear stress, $A$ is the buoy surface, and $\mathbf{r}$ is the vector between the centre of each cell surface panel and buoy's centre of gravity. As described in Section 2.1, the mooring system behaves as a linear spring exerting the so-called spring force, $F_{\mathrm{spring}}$, on the buoy. This force is implemented through the *linearSpring* restraint, available in the *sixDoFRigidBodyMotion* OpenFOAM library. As in the linear model, the spring constant is $k = 5000$ N/m. The changes in water level elevation and the slack in the mooring line, slack $= 0.64$ m, are implemented through the restraint. In particular, the mooring line rest length is defined as:

$$\text{rest length} = \text{depth} - \text{buoy draft} + \text{slack} - \text{water level} \qquad (3)$$

The dimensions of the numerical wave tank are $3\lambda \times 100$ m $\times 2d$ m (L $\times$ W $\times$ H); the length parametric changes with the wave length, $\lambda$, and the height is a function of water depth, $d$. To avoid the wave reflections, the wave tank's length downstream of the buoy and the width are determined based on the sensitivity study presented by the authors [27]. The buoy is placed $1\lambda$ from the inlet boundary, providing enough space for the wave to propagate. In general, as suggested by the ITTC Recommended Procedures and Guidelines [28], the inlet boundary should be at least six wavelengths ($6\lambda$) in front of the device. However, due to computational resource limitations, a longer numerical domain would add a significant computational cost to our simulations. Figure 4 shows the wave tank's computational mesh, which consists of the background and overset meshes. The higher resolution region close to the water surface extended $2H$ below and above the water level, is used to properly capture the wave propagation, avoiding the excessive damping. The spatial discretization is equal to 17 CPH (cells per wave height) following the mesh convergence study presented in [27], preserving the spatial discretization error <0.6%. In the area close and around the buoy, the mesh is even more refined so that the hydrodynamic forces and the turbulence phenomena are well-captured. Wall functions are applied for the boundary layer solution while the $y+ \in [30, 300]$, satisfying the conditions described in [29]. Further description of the wall functions utilized in this study are found in Table 3. The computational domain consists of $11 \times 10^6 - 15 \times 10^6$ cells (depending on the sea state), and the each simulation requires 20–30 h in a High Performance Computing (HPC) cluster utilizing 128 parallel processors.

Due computational costly CFD simulations, an irregular sea state is numerically represented by an equivalent regular wave train, and this is equal to the maximum individual wave at a given sea state. According to the recommended practices [30,31], the sea state's maximum individual wave height is approximated as $H_{\max} = 1.9H_{\mathrm{s}}$, assuming that the distribution of wave heights can be represented by the Rayleigh distribution and the sea state consists of $N = 1000$ waves:

$$H_{\max} = H_{\mathrm{s}}\sqrt{0.5\ln N} \qquad (4)$$

Table 2 presents the wave characteristics for the four examined sea states and the equivalent regular waves. Each sea state is characterized by the significant wave height, $H_{\mathrm{s}}$, and the peak period, $T_{\mathrm{p}}$, while the equivalent regular wave is described by the wave height, $H$, and wave period, $T$. The numerical wave generation and absorption is implemented using the IHFOAM toolbox. In particular, the static boundary method is applied at the *inlet* boundary for the wave generation; the inlet wave velocity and surface elevation are determined based on Stokes wave theory. The active wave absorption at the *outlet* boundary cancels the upcoming wave by applying a uniform velocity profile in the opposite direction

of the upcoming wave. For all the walls, the pressure is defined such that there is zero flux using the `fixedFluxPressure` boundary condition. However, the top of the numerical wave tank, which is the *atmosphere* boundary, allows the air to pass through and a uniform reference pressure, $p_0$, is defined using the `totalPressure` boundary condition. The `pressureInletOutletVelocity` boundary condition along the *atmosphere* indicates that velocity of the flow is calculated based on the pressure of the flow through the boundary. The `slip` wall boundary condition is imposed on the *seabed* and *side walls* boundaries and erases the normal component of the variable at the surface and keeps the tangential components unaffected. The boundary condition `zeroGradient` indicates that the gradient of the respective quantity is zero, meaning that the actual value is constant. In Table 3, the assigned boundary conditions are listed. The initial values for turbulent kinetic energy, $k$, and turbulent dissipation rate, $\omega$, are calculated based on the incoming wave celerity as suggested by [26].

**Table 3.** CFD boundary conditions.

|  | Inlet/Outlet | Seabed | Atmosphere | Side Walls | Buoy |
|---|---|---|---|---|---|
| alpha.water | waveAlpha [1]/zeroGradient | zeroGradient | inletOutlet | zeroGradient | zeroGradient |
| Pressure | fixedFluxPressure | fixedFluxPressure | totalPressure | fixedFluxPressure | fixedFluxPressure |
| Velocity | waveVelocity [1] | slip | pressureInletOutletVelocity | slip | movingWallVelocity |
| $k$ | fixedValue/inletOutlet | kqRWallFunction | inletOutlet | zeroGradient | kqRWallFunction |
| $\omega$ | fixedValue/inletOutlet | omegaWallFunction | inletOutlet | zeroGradient | omegaWallFunction |
| nut | calculated | nutkWallFunction | calculated | calculated | nutkWallFunction |
| pointDisplacement | fixedValue | calculated | calculated | calculated | calculated |

[1] IHFOAM toolbox.

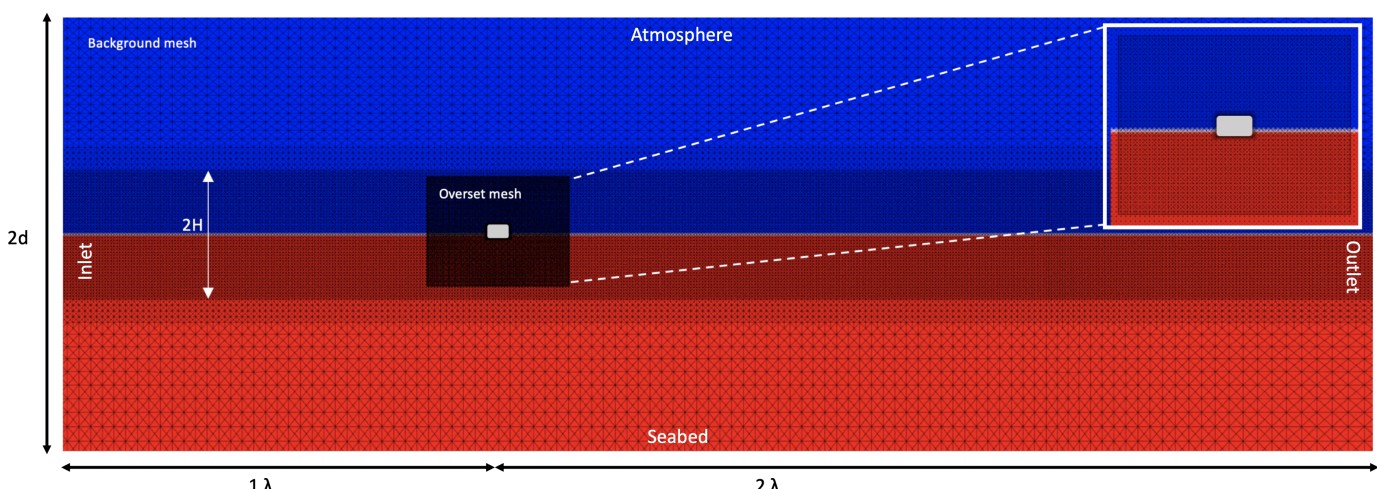

**Figure 4.** Schematic depiction of the numerical wave tank in the *xz* plane, with the dimensions and boundaries labelling. The letters *d*, *λ*, and *H* denote the water depth, wave length and wave height, respectively. The background and overset mesh regions, and the boundaries' labelling are shown. The region of higher mesh resolution close to the water surface and around the buoy captures the wave propagation and the dynamic and turbulence effects, respectively.

*2.4. Extreme Value Analysis*

A statistical analysis of the extreme force data is performed using peak-over-threshold to inspect the tail of the distribution of the forces. A certain threshold is chosen and the exceedances of the force above the chosen threshold are modelled by a generalized Pareto distribution (GPD). The purpose is to investigate the reliable extrapolation of the extreme forces. It is assumed that the force data consists of independent random observations which are in more general terms stationary, i.e., the stochastic properties of the data series are constant over time or are unfollowed by any strong pattern of variation over time. Practically, there are mutual dependencies among the variables in extreme events, i.e., perceived through the consecutive observations. For instance, in the event of an extreme

force, the data points gathered around the time of the event are most likely in a similar order of magnitude, which indicates the mutual dependency among data points. Hence, a mathematical treatment is required to address this issue. Thereby, an empirical strategy to cluster the data using a declustering technique has been applied according to [32]; and then the distribution is fitted to the maximum exceedance of each cluster [32].

Selecting one distribution over the others from the GPD family is based on the estimated statistical parameters [33]. Here, the choice of distribution for the GPD model is made by evaluating the shape parameter and its confidence interval. If zero falls in a confidence interval of the shape parameter, the exponential distribution (ED), i.e., a subset of the GPD model, would have superior inferences over a generalized formulation of Pareto distribution. This is due to simplicity and lower uncertainty in the exponential distribution. To this extent, the exponential distribution (ED) is selected for the GPD model.

In the peak over threshold method, the exceedance $z$, above a chosen threshold $u$, is modelled by a GPD with cumulative distribution function defined as [32]

$$H(z) = \begin{cases} 1 - \left(1 + \frac{kz}{\tilde{\sigma}}\right)^{-1/k}, & k \neq 0 \\ 1 - \exp\left(-\frac{z}{\sigma}\right), & k = 0 \end{cases} \tag{5}$$

where $\tilde{\sigma}$ is a scale parameter and k is a shape parameter for $\{z : z > 0 \text{ and } 1 + kz/\tilde{\sigma} > 0\}$, while $-\infty < \mu < \infty$, $\sigma > 0$, and $-\infty < k < \infty$. The special case of $k = 0$ corresponds to the exponential distribution. The mean of the exceedances that are modelled with GPD is given by

$$E(u) = \frac{\tilde{\sigma} + ku}{1 - k}. \tag{6}$$

The choice of threshold is made with the aid of a mean residual life plot, i.e., extracted from the measured data, and shape parameter stability plot for GPD model. The former shows the mean of the exceedances across a range of thresholds, and this estimate is expected to change linearly in the threshold range in which the Pareto distribution is a suitable model, according to Equation (6). In the latter, the shape parameter is expected to be constant above threshold $u_0$, where the asymptotic assumption for Pareto model is valid [32].

In the declustering technique, it is important that the cluster size is neither too small to not refine the mutual dependency, nor too large to miss the series of observations that could have been considered as independent, and this choice is a trade-off between the variance and bias [32]. Here, *run time* specifies the size of the cluster. When a consecutive number of observations fall below the threshold for the period of time equal to the run time, the cluster is terminated. The assessment of the GPD model while satisfying the terms of declustering method is not straightforward, since it involves optimizing both the choice of threshold and the run time. Therefore, the sensitivity of the model has been evaluated through two sets of optimization by varying the run time for different values of threshold and computing the standard error and return period to provide an impression of the stability of the GPD model. Additionally, the value chosen for the threshold from the optimization can be cross-checked with the mean residual life plot.

The validation of extrapolating the extreme values by GPD with the special case of the ED model has been appraised through four diagnostic plots of probability, quantile, return level, and density. The probability and quantile plots should illustrate linearity with respect to the diagonal line, and the return level plot should display proper compliance between the empirical data and the model in terms of return period. In the extreme value analysis, the agreement between the empirical data and the model is specifically important for high force data points. In the end, the density plot shows a comparison between the probability density function of the fitted model and histogram of the data. In a general sense, the density function plot is less informative compared to others since the histogram can easily revolve depending on the choice of the grouping intervals [32].

### 3. Results

*3.1. Offshore Experiments*

Two periods of the measurement series, first days and the last day, experienced waves that were normal or higher than normal. During the last day, just before the cable broke, there was a period with forces close to 2000 N and the highest measured force was 1989 N, see Figure 5. The highest force measured for each sea state are shown as a function of significant wave height in Figure 6.

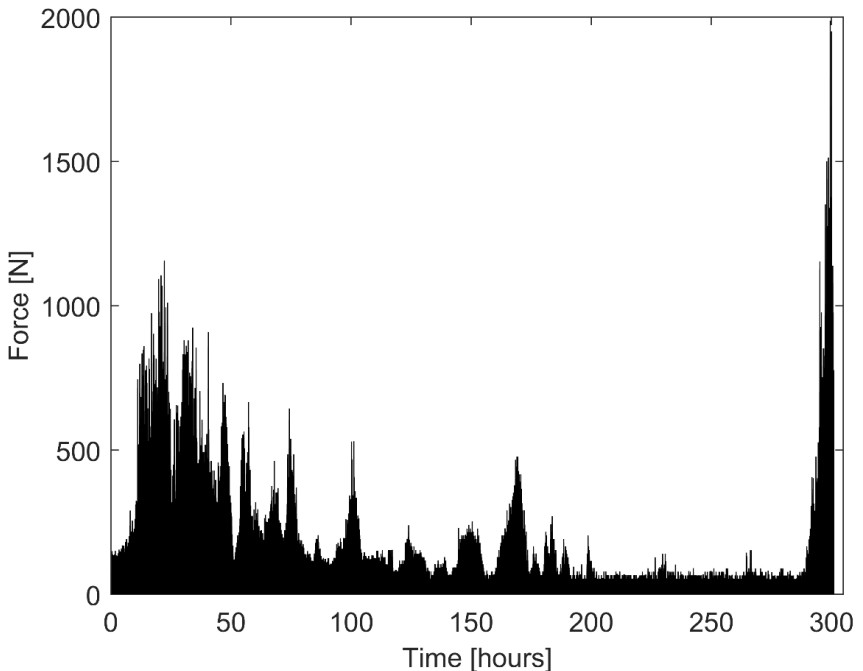

**Figure 5.** Force data from the load cell during the whole measurement period.

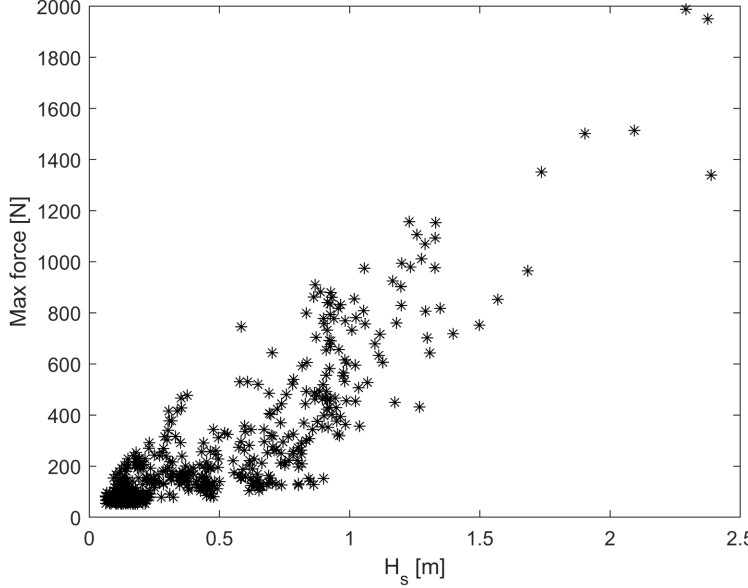

**Figure 6.** The maximum force for each sea state are plotted versus the significant wave height.

Data shows a trend with higher max forces for higher wave heights, see Figure 6. However, there are also several deviations from the pattern. Sea states with the same significant wave height show huge differences in measured forces.

### 3.2. Extreme Loads and Loads over Threshold Analysis

Figures 7 and 8 present the mean residual and parameter stability plot for different thresholds of the forces. The mean residual life plot, Figure 7, shows a linear and constant pattern for thresholds between 880 N and 1120 N, while the shape parameter slightly varies around zero in this range. The lower value of this range is selected as the initial value for the threshold. The fact that zero lies in the shape parameter confidence interval in the range of the threshold where GPD is valid implies the necessity to check the exponential distribution model whose inference would be preferred due to its simplicity, which introduces lower uncertainty in the model. In this case, Equation (6) suggests that the mean residual value should be equal to the scale parameter for this range of the threshold. The diagnostic plots for the ED model without declustering is displayed in Figure 9, which shows a relatively good agreement between the empirical data and the model for extrapolation of extreme forces.

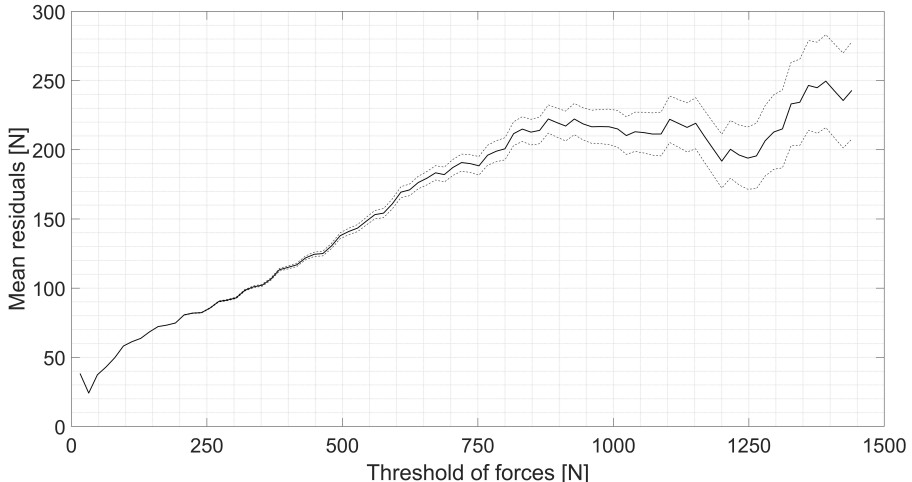

**Figure 7.** Mean residual plot with 95% confidence interval.

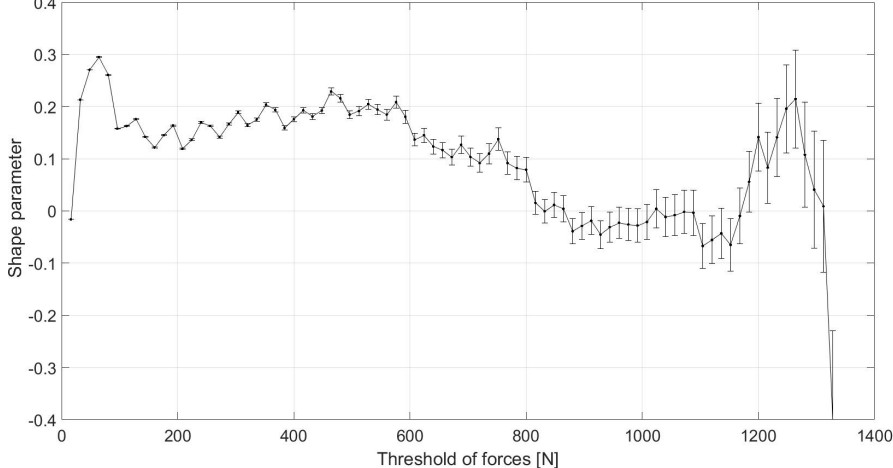

**Figure 8.** Stability plot of the shape parameter of the GPD. The shape parameter versus threshold together with the 95% confidence interval of the parameter.

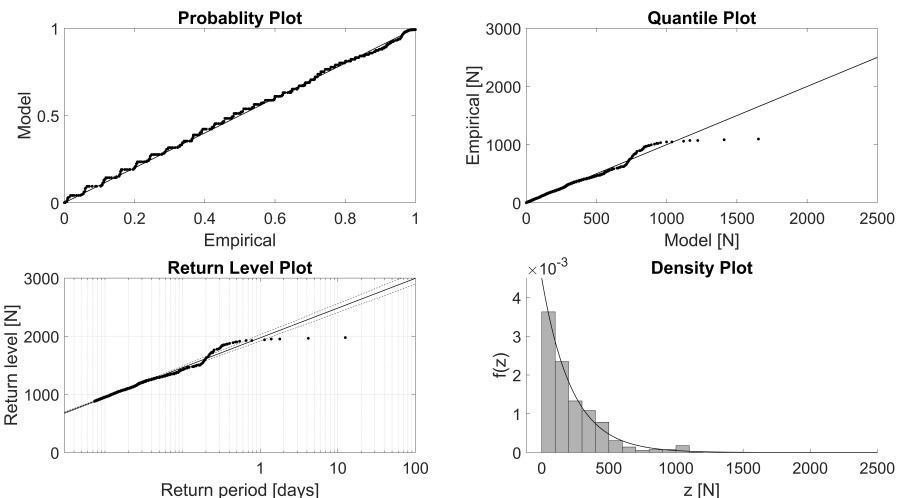

**Figure 9.** Diagnostic plots for ED fit to the force data with the threshold of 880 N without declustering. In the return level plot, the dashed lines show the lower and upper bound of 95% confidence interval. In the density plot, *z* corresponds to forces above threshold (exceedances).

The quantile and probability plots offer an adequately linear trend for low force data points with relatively low return level error. However, for the high force values, the quantile plot and the return level plot show clear disagreement between the empirical data and the model. One reason can be due to the temporal dependency of the exceedances, which can be treated by using the declustering technique. In the declustering technique, the threshold and the run time are further optimized with the aim of selecting the threshold with higher accuracy in the selected range of 880 N to 1120 N. The optimization goal is to minimize both threshold and run-time to achieve minimum error while providing a reasonable return period for high forces. The error for the return level decreases with lower threshold values, and the return period increases with increasing the run-time in all threshold values. The optimization resulted in the threshold of 880 N and run time of 0.0134 h.

Figure 10 shows the diagnostic plots for the fitted ED with the optimized threshold and run-time. This model shows clear improvement when compared with empirical data where superior agreement at high force data points can be seen for both the quantile and probability plot. Thanks to the simplicity of the exponential distribution, the confidence interval provides a narrow window for high forces. The scale parameter for the exponential model is obtained as 203.04, which is close to the mean residual value in the range of the threshold where the GPD is a valid model.

Following the satisfactory results obtained with this model, the extreme value data is further extrapolated over 8.94 years, which corresponds to 20 years for a full-scale system, for the calm period of the year. This period compromises roughly five months of the year between April to August according to [21] in the North Sea. The return level for 8.94 years is 2656 N with a 95% confidence interval of [2200, 3112] N, as shown in Figure 11.

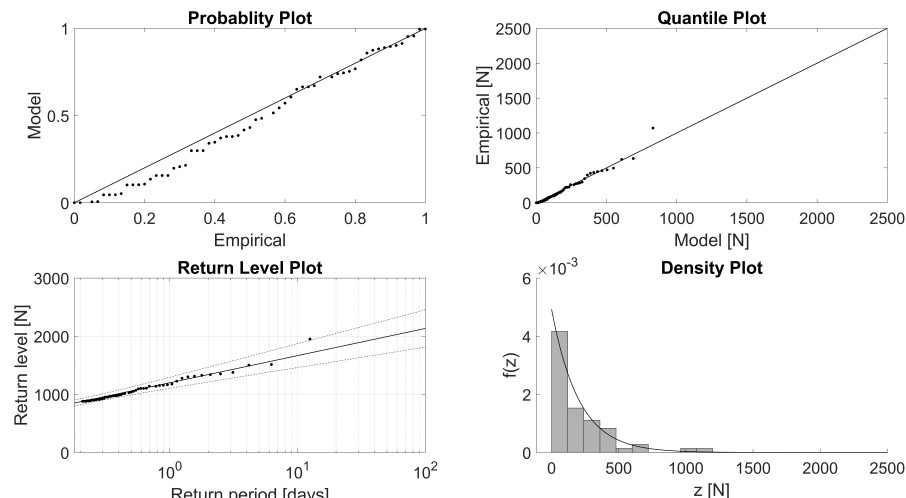

**Figure 10.** Diagnostic plots for ED fit to the force data with the threshold of 880 N and run time of 0.0134 h. In the return level plot, the dashed lines show the lower and upper bound of the 95% confidence interval. In the density plot, $z$ corresponds to maximum force of each cluster of exceedances.

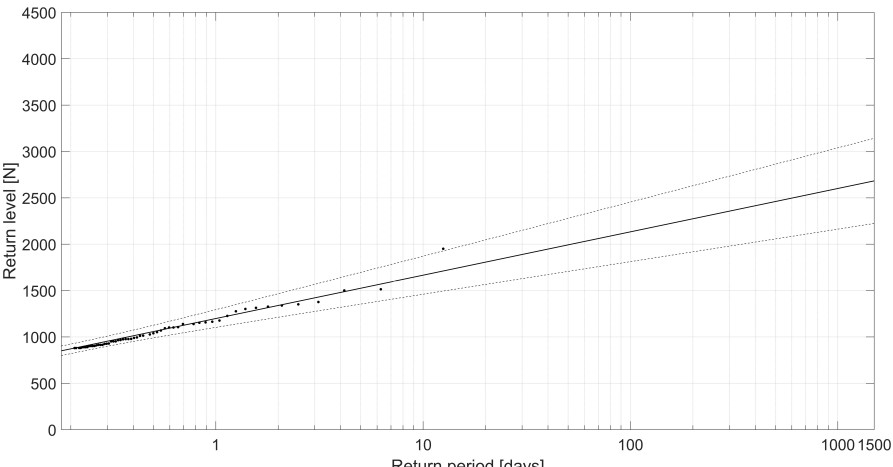

**Figure 11.** The return level versus return period for extrapolated exponential model for the calm period of the year over 8.94 years (i.e., corresponding to about 20 years for a full-scale system). The dashed lines show the lower and upper bound of the 95% confidence interval.

### 3.3. Comparison to Numerical Simulations

As the linear model assumes only heave motion and accounts for the slack mooring line (resulting from the line being 0.64 m longer than the water depth) simply by activating the spring force only if the vertical buoy position is greater than 0.64 m, we expect that the model should give zero line forces in sea states where the waves are too low to stretch the mooring line to a non-slack position. In reality, this does not happen, as the buoy will drift in surge and sway, avoiding slack lines even in low-amplitude waves. With this insight, the linear model is not expected to represent the physical system accurately in sea states with low energy—in the model, the waves are too low to simulate a non-slack line and obtain a non-zero line force. On the other side, for large and steep waves, non-linear phenomena are expected, and the linear model is not expected to provide accurate predictions.

The obtained results from the high-fidelity CFD model for sea states #1–4 are shown in Figure 12. The buoy response in heave and surge is shown, as well as the line force during the same time. As the CFD simulations are carried out not in irregular waves with significant wave height $H_s$, but in equivalent regular waves with wave height $H = 1.9H_s$,

the dynamics are expected to follow a regular wave pattern, which is seen in the figures. After the first waves have activated the buoy into oscillation, the motion and line force should follow a (more or less) stable pattern that is repeated for each wave period. The maximum force should, therefore, be taken as the average over the maximum force obtained in each regular wave period after the motion has been stabilized [33]. The average value of the peak force is estimated based on the last 4 wave cycles, as shown in Figure 12. In the force subfigure, the dashed line indicates the window in which the maximum force is averaged.

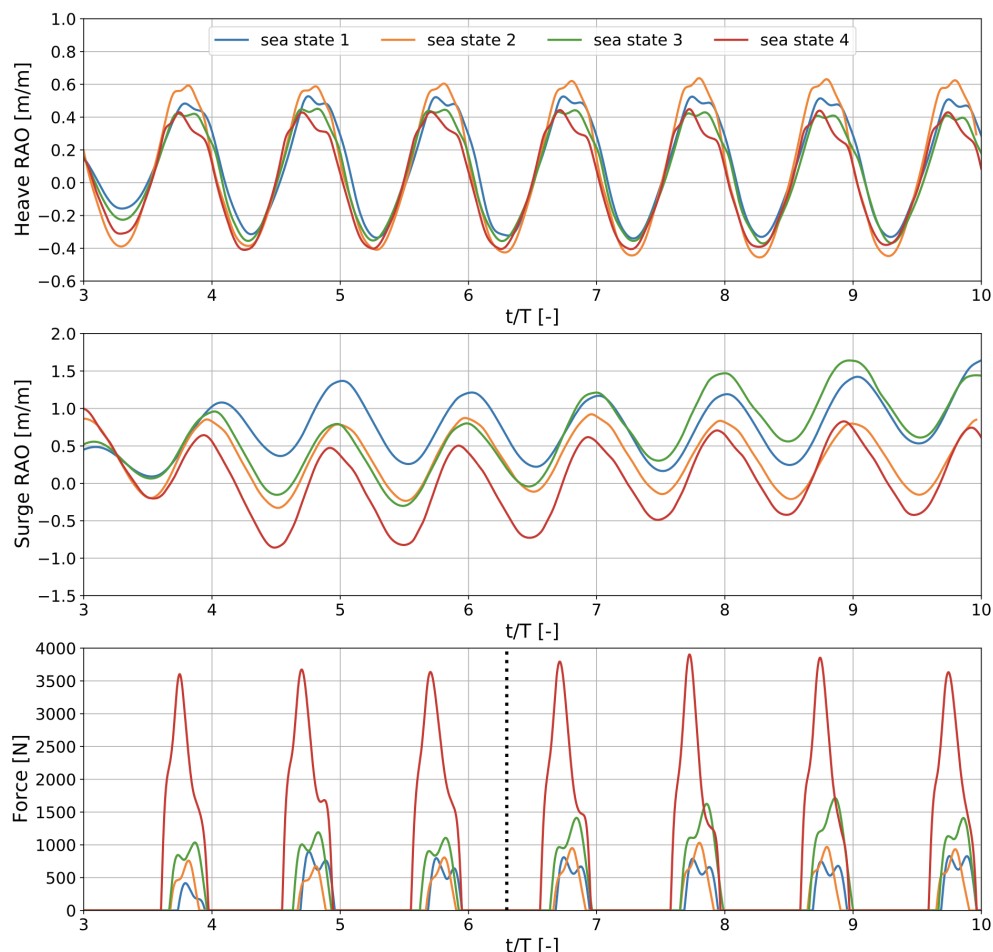

**Figure 12.** Results from CFD simulations of sea state #1–4.

In Figure 13, the computed maximal forces using both the linear model and the CFD model are compared to experimental data for the four sea states given in Table 2. As can be seen from the figure, the maximal force obtained by the CFD simulations correspond well to the experimentally obtained values for the two sea states with lowest wave height. For the two sea states with higher waves, sea state #3 and #4, the agreement between the experimental value and model value deviates a lot. For both numerical models, the experienced line force increases with increased wave height. However, this behaviour is not really seen for the experimentally measured force. This can be explained by looking back at Figure 6, where it is clear that the maximal force increases with increasing significant wave height in general, but that the scatter in the experimental data is large, so this trend is not expected to be visible when comparing a few individual sea states.

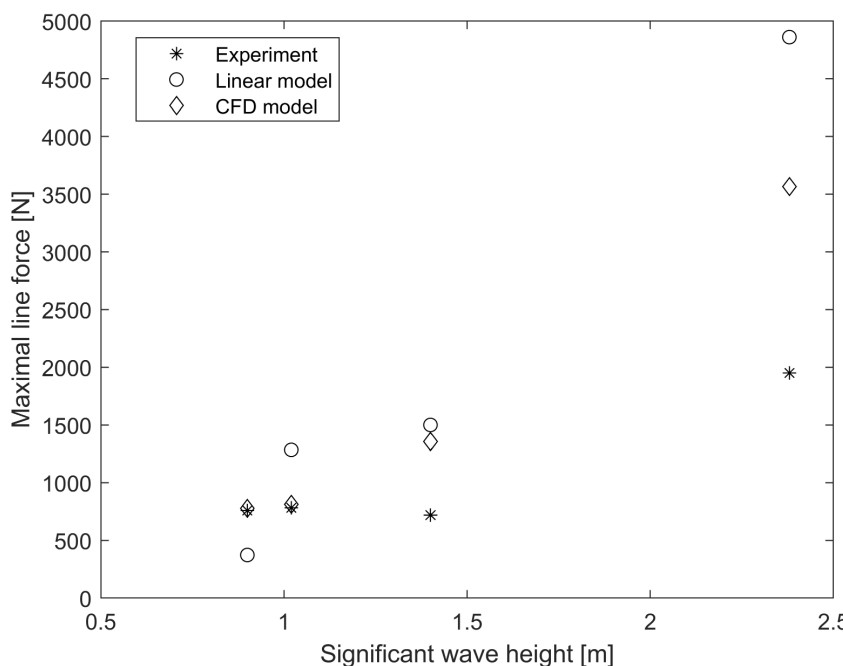

**Figure 13.** Comparison between predictions from the approximate linear model, the high-fidelity CFD model, and experimental data, for all sea states given in Table 2.

## 4. Discussion

The main objective with this project is to gather force data during storms that can be converted to full scale values that give an understanding of the loading on wave energy converters. A first prototype was installed in November 2017 and was indeed subjected to the biggest storm during the last ten years. As a result, that buoy was destroyed by the storm, which indicates the severity of the harsh offshore wave conditions. In this second trial, the ambition was lowered and the aim was to see if the system works. It was therefore deployed during the usually calm period of May. It was a calm period, but in the end of the period, the buoy collected data from waves that were larger than average conditions. However, data from extreme waves was never collected.

The site is located in a near-shore environment with a water depth somewhere between deep water and intermediate water conditions and surrounded by small islands and islets. It is therefore a rather complex physical environment with a combination of North sea-type sea states and locally produced/affected wave components, depending on the wind direction. Most sea states are therefore a mixture of developing and developed sea states.

The two numerical models used in this paper both have their advantages and drawbacks. The linear potential flow model is based on crude approximations of the fluid (ideal, incompressible, irrotational fluid with low amplitude waves) and the buoy dynamics (restricted to heave motion and a stiff connection line), and the results are therefore not expected to give accurate predictions of the maximal forces. However, the method is very fast, so that one full sea state computes within seconds without parallel processors on a standard PC, and thus all sea states for the full duration of the test period can be modelled without problem. In Figure 13, it can be seen that the linear model gives values that are in the same order of magnitude as the experimental value, but no accurate prediction is obtained. This can be expected since the linear model is indeed a very crude representation of the experimental set up. It should be clarified that the scope of the paper is not to validate the numerical models; but since linear models are often used due to their simplicity and efficiency, there is a value in investigating how useful such a model is in estimating the line force when actual offshore data are available. That is our motivation.

On the other side of the computational spectrum, the CFD model includes viscosity and turbulence, solves the motion in six degrees of freedom, and the mooring line force is zero only when the line is slack. In other words, the model is a much more realistic representation of the physical system. The well-known drawback, however, is the high computational cost. In the current work, a limited number of CFD simulations is available due to computational resources; therefore, only a few sea states are studied. In addition to the CFD computational limitations, the irregular sea state is numerically represented as an equivalent regular wave, and the wave–structure interaction is simulated for a few wave cycles. This approach still requires over 24 h of computation on 128 parallel processors at a High Performance Computing (HPC) cluster. However, the regular wave representation includes uncertainty for the CFD model in terms of how accurately the extreme forces can be captured. Another uncertainty is related with the choice of the equivalent wave height, which is estimated as $H = 1.9H_s$. This is a vague approximation based on the assumption that the Rayleigh distribution describes the wave height and the sea state consists of 1000 waves. In our case, the sea states last for 67 min (in full scale) instead of the 3 h typical duration, therefore, the number of waves might be smaller. In addition, the maximum individual wave height can be more accurately defined by fitting the proper distribution in the available data. Thus, a somewhat smaller maximal wave height than $1.9H_s$ could be expected in each sea state. We could also observe from the data recorded during the experimental campaign that the maximum individual wave height at some sea states were lower than the $1.9H_s$ relation gives. Another large uncertainty in the numerical model is how the rubber damper and mooring line are represented. The line and damper are modelled as a linear spring in the CFD model, while in reality the system is more complex.

As can be seen in Figure 13, the CFD model predicts the force better than the linear model; however, both models fail to give a realistic prediction in the higher wave sea states. As discussed, the failure of the numerical models in these cases can be acknowledged to the large uncertainties present in the physical model. The simple, yet important conclusion to be drawn is that numerical simulations, not even high-fidelity ones, can be expected to give accurate predictions if they are not validated to be exact representations of the real system, including the uncertainties involved.

The extreme values of force data are extrapolated for longer periods of time using the statistical model peak-over-threshold. A GPD distribution is fitted to the force data above a certain threshold, a relatively good agreement between the model and the empirical data is obtained. The declustering technique is further employed to remove the temporal dependency in the data and improve the return level by slightly increasing the variance of the prediction. The exponential model with a threshold of 880 N and a run time of 0.0134 h is considered for this analysis. It is worth mentioning that extrapolating the extreme values to relatively large time periods should be handled with caution because the short measurement period of two weeks adds uncertainty to the model in large return periods and normally a measurement period of a couple of years is required to be able to predict long return periods such as hundred years. Furthermore, according to [12], the calm climate period of the year follows from April to August, and the probability of occurrence of larger force is higher in the rest of the year. Since the two weeks force measurement were carried out in May, the predicted return period in this extreme value analysis is mostly in agreement with the calm climate period of the year. Hence, as a point of interest, the exponential model is further extrapolated for the calm period of the year, where a return level of 2656 N is achieved for the return period of 8.94 years.

A direct comparison between any numerical model to the experimental data is inevitably connected to large uncertainties. Not only are there approximations and uncertainties in the numerical models, but also in the experimentally data obtained. One uncertainty stems from the fact that the wave data is obtained from a wave measurement buoy not in the direct vicinity of the force measurement buoy, but 100 m away from it. Another uncertainty is the line length; the mooring line was not completely stretched, but instead deployed with a slack of approximately 0.64 m. This additional line length has been incor-

porated in the numerical simulations, but the results are rather sensitive to the value of the additional length—if, for example, a slightly longer line length had been defined in the numerical simulations, lower computed maximal forces would have been obtained. This experimental set up is relatively simple with few mechanical parts and a rubber spring that gives rather smooth dynamics. A full scale WEC with a mechanical end stop would increase the number of unknowns and thus further increase the difficulties to replicate its dynamics with a numerical model. Another factor is the individual wave steepness of each wave component. Wave steepness was first included in the analysis of the four sea states, but no trend could be visible. Nevertheless, it is probably a contributing factor.

A large difference in the measured force for waves with the same significant wave height can be seen. Probably the biggest uncertainty lies in the energy source itself and its interaction with the device. The unique motion history of the buoy is as important as the magnitude of the wave. Capturing this during long-term offshore conditions is a big challenge—both for low- and high-fidelity methods.

One obvious improvement is to gather much more data; this is after all just two weeks of data. More data will make it possible to see patterns that can reveal shortcomings of the models. The same regards CFD simulations; a more detailed representation of the waves and the mechanical system as well as simulations for more sea states would give more insights. Various data-driven methods such as machine learning algorithms can open possibilities for faster simulations and could increase the accuracy of the modelling since they can adapt to the varying environment and reveal hidden patterns.

## 5. Conclusions

A force measurement buoy in the scale 1:5 to the Uppsala University WEC has been built and installed for one month at Uppsala University's research test site off the west coast of Sweden, and collected force data for about two weeks. This paper presents data of line force measured in an offshore environment, and thus contributes with new knowledge to the very limited amount of published data from offshore experiments. This data provides important insights into how WECs should be designed to survive harsh wave conditions in realistic offshore environments. The main conclusions are that

(a)    Forces up to 2 kN were observed;

(b)    Even for waves of similar significant wave height, a large variance in maximum force was measured.

The peak-over-threshold model with exponential distribution resulted in a satisfactory fit to the empirical data. By further extrapolating the exponential model, a return level of 2656 N is predicted for the calm period of the year over 8.94 years, which corresponds to the return level of 332 kN with a return period of 20 years for a full-scale system.

Data have been compared to two different types of numerical models, one low fidelity linear wave theory model and one high fidelity CFD model. As expected, the CFD model shows the best agreement between the predicted force values and the measured ones, but the linear model predicts values that are within the same order of magnitude and could thus be used as a fast method to indicate which forces could be expected. In most cases, both models predicted forces that are higher than those measured. This can be attributed to the unavoidable differences between the real system and the numerical models, and to the uncertainties involved in the experimental setup. In particular, inevitable simplifications in the numerical models, distance between the force measurement buoy and wave measurement buoy, the relation $H = 1.9H_s$ that was used to generate the regular waves in the CFD simulations, and the unique motion history of the buoy.

The huge difference in the measured force for sea states with the same significant wave heights indicates that it will be difficult to predict the force to a high accuracy based on the significant wave height. This points out the importance of conducting offshore experiments and not simply relying on numerical simulations or even controlled tank experiments.



**Author Contributions:** Designed the experiments, J.E. and M.G.; conducted the experiments, J.E.; processed the experimental data, J.E.; analysed the data, J.E. conducted the extreme value analysis, Z.S.; conducted the CFD modelling, E.K.; conducted the linear modelling, C.S.; discussed the results, all authors; writing and editing, all authors; supervision, J.E. and M.G.; funding acquisition, J.E. and M.G. All authors have read and agreed to the published version of the manuscript.

**Funding:** The authors would like to thank The Swedish Energy Agency for funding this project (Project no. 47264-1) within the national Swedish research program for marine energy conversion. The research in this paper was also supported by the Centre of Natural Hazards and Disaster Science (CNDS), Sweden, the Onassis Foundation, scholarship ID: F ZP 021-1/2019-2020, Swedish Research Council (grant 2015-04657), the Lars Hiertas Foundation, Wallenius Foundation, and the Bengt Ingeströms scholarship.

**Acknowledgments:** The CFD simulations were performed on resources provided by the Swedish National Infrastructure for Computing (SNIC) at HPC cluster Tetralith. The Swedish STandUP for Energy research alliance, a collaboration initiative financed by the Swedish government, is acknowledged for contributing to the research infrastructure.

**Conflicts of Interest:** The authors declare no conflict of interest.

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
