# Peer review of "Offshore Measurements and Numerical Validation of the Mooring Forces on a 1:5 Scale Buoy"

_jmse, doi:10.3390/jmse11010231_

Round 1
Reviewer 1 Report
See attached file for comments

Reviewer 2 Report
The submitted manuscript presents results from field measurements of mooring forces on a buoy deployed at Uppsala University’s test site for wave energy research at the west coast of Sweden. In addition, the manuscript presents the validation of two numerical models using the recorded data. The topic fits the scope of JMSE; however minor revision are required to render the manuscript fit for publication. In the following, general and specific comments are provided which need to be addressed in a revised version of the manuscript, prior to publication.
General comments
================
(i) The authors state that the acquisition of "actual offshore data" is the novelty presented in the paper. The reviewer agrees with this statement; however, in my opinion, the presented (brief!) literature review does not clearly highlight the need for "close-to-full-scale" data. The author's are asked to strengthen the literature review (e.g. I am missing references to the work done by Jacobsen on the large scale WaveStar device) and highlight more clearly the specific motivation of the study.
(ii) In the reviewers opinion, the focus of the paper is slightly unclear. The authors present both in-situ data and the analysis thereof, as well as numerical modelling approaches; however, none of these two parts are handle in very great detail, giving the reader a hard time to clearly see the added value of the paper. Ending on the note that offshore experiments are important is, in the reviewer's opinion a bit weak and, based on the data at hand, more valuable conclusions could be drawn. It would, in my opinion, be interesting to reflect on the possibility of jointly using numerical and in-situ data. Can the authors comment on the potential of having data from these two approaches at hand?
(iii) It would be very valuable, if the in-situ data would be made publicly available. Is this planned? Please comment.
Specific comments
=================
(1) Line 4: Please specify what is meant by a "force measurement buoy"
(2) Line 8: Similar to the above. Which "force data" is meant here
(3) Line 11: Since you are referring to field measurements, I would not refer to "experimental [...] uncertainties". Please rephrase
(4) Line 46: Rephrase "huge" to "significant"
(5) Line 61: "The data" What is meant here? Please provide more details
(6) Please revise and merge lines 74-77
(7) Line 86: Is is stated that the buoy is "equipped with 20 kg ballast to equalize the mass of the battery". In my opinion, this does not make sense. How can additional ballast "equalize" the mass of the battery? Do you mean in terms of mass distribution? Please clarify
(8) From Fig. 1 it is not clear to me, if the load cell is always in line with the mooring line or if it is fixed statically to the buoy. Please clarify.
(9) I am missing information on the inertia of the buoy. This is critical for the validation of high-fidelity models
(10) I am missing a figure of the geographical location of the test site (e.g. GIS based). Please also add a typical scatter diagram of the test site
(11) Line 130: "the buoy is assumed to move only in heave"; were motion data of the buoy recorded? How valid is this assumption? Please comment
(12) Section 2.3.1: What kind of approximation method is used for the convolution integral?
(13) Line 156ff.: "using the volume of fluid phase-fraction based on interface capturing approach" This sentence is odd, please rephrase/correct
(14) Line 182ff: "This force is implemented through the linearSpringEndStop restraint, available in the sixDoFRigidBodyMotion OpenFOAM library" Please provide a reference to the restraint. As far as I am aware, this restraint is not readily available in OpenFOAM
(15) Figure 4: Please provide a close-up of the boundary layer mesh; Please also provide the dimensions of the domain in the figure
(16) Line 199: "Wall functions"; Please provide more details on the specific wall functions used
(17) Table 3: The definitions of the boundary conditions are very OpenFOAM-specific and not self-explanatory for non-OpenFOAM users. Please provide general definitions of the boundary conditions
(18) Line 269f: "Two periods of the measurement series, first days and the last day, experienced waves that were normal or higher than normal." I'm not sure if I can follow this statement, please clarify
(19) Figure 9 & 10 & 12: The font size is too small in these figures
(20) Line 324: The response in pitch is not shown in Figure 12; also, no identifiers (a)-(d) are given
(21) Figure 12: Why is data for sea state 2 only shown until approx. 9 t/T? Why is data only shown for t/T>3?
(22) Line 331f: "The average value of the peak force is estimated based on the last wave cycles as shown in Figure 12(d)." I can't follow this. Please provide an exact interrogation window length based on t/T and highlight this in Fig. 12
(23) Lines 334: Please provide quantification of the deviation between the different data points
(24): Line 342f: "but that the scatter in the experimental data is large, so this trend is not expected to be visible when comparing a few individual sea states" Would it not be possible to add standard deviations/error bars in Fig. 13 in order to highlight the spread in the experimental data in the figure? Please comment
(25) Please comment on the specific selection of the sea state for the numerical models. Looking at Fig. 13,the selection at the lower Hs does not appear meaningful, if the difference in the experimental data is insignificant
(26) Figure 13: Please comment on the fact, that the lin. model underestimates CFD at the lowest Hs, this is unexpected.
(27) Line 366f: "And as seen in Figure 13, the results are relatively close to the experimental results for some sea states, but are far from accurate for others." This is a relatively meaningless statement. Please provide a more in-depth discussion.
(28) Line 395ff: "The simple, yet important conclusion to be drawn is that numerical simulations, not even high-fidelity ones, can be expected to give accurate predictions if they are not validated to be exact representations of the real system, including the uncertainties involved." In the reviewer's opinion this is indeed an important conclusions; however, to fully support this, the paper should discuss more clearly the uncertainties and, in particular, method/procedure, how to possibly mitigate those. The reviewer appreciates the list of uncertainties in lines 418ff.; however mitigation measures should be discussed
(29) Line 464f: "... will be difficult to predict the force to a high accuracy based on the significant wave height, no matter how sophisticated model one uses." The reviewer disagrees with this statement, if e.g. more "sophisticated" mooring models would be used or irregular sea states would be considered, I would argue that the results would be better. Hence, "no matter how sophisticated" is not a reasonable statement.
